# Cytoskeletal Protein Variants Driving Atrial Fibrillation: Potential Mechanisms of Action

**DOI:** 10.3390/cells11030416

**Published:** 2022-01-25

**Authors:** Stan W. van Wijk, Wei Su, Leonoor F. J. M. Wijdeveld, Kennedy S. Ramos, Bianca J. J. M. Brundel

**Affiliations:** Department of Physiology, Amsterdam Cardiovascular Sciences, Amsterdam University Medical Centers, Vrije Universiteit Amsterdam, 1081 HV Amsterdam, The Netherlands; s.w.vanwijk@amsterdamumc.nl (S.W.v.W.); w.su@amsterdamumc.nl (W.S.); l.f.j.wijdeveld@amsterdamumc.nl (L.F.J.M.W.); k.silvaramos@amsterdamumc.nl (K.S.R.)

**Keywords:** atrial fibrillation, genetics, cytoskeletal proteins, cardiomyocytes, DNA damage

## Abstract

The most common clinical tachyarrhythmia, atrial fibrillation (AF), is present in 1–2% of the population. Although common risk factors, including hypertension, diabetes, and obesity, frequently underlie AF onset, it has been recognized that in 15% of the AF population, AF is familial. In these families, genome and exome sequencing techniques identified variants in the non-coding genome (i.e., variant regulatory elements), genes encoding ion channels, as well as genes encoding cytoskeletal (-associated) proteins. Cytoskeletal protein variants include variants in desmin, lamin A/C, titin, myosin heavy and light chain, junctophilin, nucleoporin, nesprin, and filamin C. These cytoskeletal protein variants have a strong association with the development of cardiomyopathy. Interestingly, AF onset is often represented as the initial manifestation of cardiac disease, sometimes even preceding cardiomyopathy by several years. Although emerging research findings reveal cytoskeletal protein variants to disrupt the cardiomyocyte structure and trigger DNA damage, exploration of the pathophysiological mechanisms of genetic AF is still in its infancy. In this review, we provide an overview of cytoskeletal (-associated) gene variants that relate to genetic AF and highlight potential pathophysiological pathways that drive this arrhythmia.

## 1. Introduction

Atrial fibrillation (AF) is the most common age-related cardiac arrhythmia in Western society [1]. AF is characterized by irregular and often very fast contractions of the atrial cardiomyocytes, resulting in an irregular heart rate, palpitations, dizziness, shortness of breath, and tiredness in the patient. AF can occur when abnormal electrical impulses suddenly start firing in the atria and override the heart’s natural pacemaker, which can no longer control the rhythm of the heart [1]. Importantly, AF is associated with severe complications, such as thromboembolic events, heart failure, cognitive impairment, and increased mortality [1]. Although in most cases, AF initially presents as short, self-terminating episodes, it often progresses into long-lasting episodes that are more difficult to reverse to sinus rhythm [1]. The progressive stages of AF are associated with structural changes that promote contractile dysfunction and the impairment of electrical conduction in the atrial myocardium [2,3,4,5,6,7]. Thus, the early detection of AF and the identification of patients at risk is of utmost importance in order to treat this arrhythmia and prevent its progression. Therefore, knowledge on the root causes of AF is essential.

Recent research investigated the potential root causes for AF. These include environmentally-induced ‘wear and tear’ AF, congenital AF, and genetic AF [1]. ‘Wear and tear’ AF is associated with the sequela of aging as well as Western diet and lifestyle-related risk factors, such as hypertension, diabetes, obesity, and coronary artery diseases, as well as various non-cardiovascular diseases, including chronic kidney disease [1]. Furthermore, an estimated prevalence of ~5% of the patients with a congenital heart disease develop AF due to a combination of flaws in embryogenesis and peri- and post-operative factors related to correction of the heart defect [8]. This so-called congenital AF is characterized by AF onset at a younger age, and these patients often rapidly progress from persistent to permanent AF [9,10]. However, not all AF patients present with predisposing ‘wear and tear’ or congenital AF. In a subset of patients who account for approximately 15% of the AF patient population, AF is familial, suggesting a genetic predisposition (Figure 1) [11,12,13,14].

Emerging research findings indicate a prominent role of genetic variations mainly in ion channel and cytoskeletal (-associated) genes in driving AF [15]. Although a research paper described the role of ion channel gene variants to underlie arrhythmias, including AF [16], insights into the role of cytoskeletal (-associated) protein variants as triggers for AF are in their infancy. In this review, we provide an overview of variants in cytoskeletal proteins associated with AF promotion and highlight potential pathophysiological pathways.

## 2. Key Role of the Cytoskeletal Network in Cardiomyocyte Function

It was only recently that the crucial importance of the cytoskeleton function to maintain balanced protein (i.e., proteostasis [3,6,17,18]) and cardiomyocyte function has been recognized. In cardiomyocytes, the cytoskeleton not only provides a communication highway by transporting proteins throughout the cell, but it also capacitates its contractile function [19,20,21]. In cardiomyocytes, the cytoskeleton is highly specialized, consisting of actin filaments, desmin (intermediate) filaments, and microtubules, interacting with membrane-associated proteins, sarcomeric and nuclear proteins, and proteins of the intercalated disk. This complex network, which also interacts with various organelles, including sarcoplasmic reticulum (SR), mitochondria, and the nucleus, plays an important role in the transmission of signals and the transport of (ubiquitinated) proteins within the proteostasis network [6,19,22] (Figure 2). Within the proteostasis network, especially the microtubules are of vital importance. As the microtubules, SR/endoplasmic reticulum (ER), sarcomeres, and mitochondria are in contact with each other, loss in contact results in Ca^2+^ overload in the organelles, unfolded protein response in the ER, and, consequently, excessive autophagic protein degradation and mitochondrial dysfunction [5,6,23,24,25]. Contact between organelles and trafficking through the cells is mediated by acetylated microtubules [3,6,26,27]. Consequently, AF-induced histone deacetylase 6 (HDAC6) activation and the subsequent deacetylation and degradation of the microtubule network have detrimental effects on the trafficking of proteins as well as on the contractile function of the atrial cardiomyocytes [3,28,29]. Therefore, a functional cytoskeletal network underlies a balanced communication within the proteostasis network and ensures proper contractile function. The conservation of this network is of utmost importance to ensure proper cardiac function. As such, attenuation of the disruption of the cytoskeleton protects against cardiac diseases such as AF and heart failure [3,26,30].

## 3. Cytoskeletal (-Associated) Variants Associated with Clinical AF

In approximately 15% of AF patients, AF occurs in the absence of common risk factors and at a younger age [1]. In these patients, AF may be familial, suggesting a heritable genetic predisposition. To explore the role of gene variations in AF, the emergence of exome and genome sequencing data has provided extensive new data and revealed a previously unsuspected link between AF and several cytoskeletal (-associated) proteins (Table 1).

Recently, several AF families have been identified to carry a mutation in genes encoding the intermediate filament proteins lamin A/C (*LMNA*), desmin (*DES*), and titin (*TTN*) [31,32,33,34] (Table 1). Intermediate filament proteins integrate the outer cell membrane via desmoplakin (*DSP*), with sarcomeric proteins such as titin, Z-disk, and the nuclear membrane, thereby regulating the sarcomere architecture and function as well as the nuclear morphology, DNA stability, and gene expression (Figure 2) [6,35,36,37]. Variants in these cytoskeletal proteins are known to be associated with the development of dilated cardiomyopathy (DCM), hypertrophic cardiomyopathy (HCM), and peripartum cardiomyopathy (PPCM) [38,39,40,41]. Of note, various studies have revealed that early-onset AF often represents the initial manifestation of the cardiac phenotype, sometimes even preceding cardiomyopathy by several years, and in the absence of common risk factors as well as gross structural changes in the heart. This observation indicates that AF is a direct consequence of the mutation rather than being caused by structural abnormalities in the ventricle [42,43]. Over 50% of *LMNA* [31,42,44], 60% of *DES* [32] and 30–60% of *TTN* [34] mutant carriers develop cardiac conduction disorders, arrhythmias, and atrial tachycardia, including AF. Furthermore, these patients reveal a more progressive form of cardiac disease with poor outcomes. These findings are further supported by other studies. In a Chinese family, four subjects revealed an AV block, and three of them suffered from AF, which was related to a new frameshift insertion in the *LMNA* gene (c.825_826insCAGG) [45]. In addition, in a study in Italian families with *LMNA* variants [46], a total of 30 subjects were included, of which 19 were positive and 11 were negative for *LMNA* variants. Of the 19 positive study subjects, 11 showed early AF versus none of the 11 negative subjects. These studies reveal a clear association between *LMNA* variants and the development of AF.

Interestingly, a recent study including early-onset familial AF patients uncovered the phosphodiesterase-4D-interacting-protein (PDE4DIP) p.A123T mutation as a genetic modifier of *DES* p.S13F [47]. PDE4DIP p.A123T increases the penetrance of cardiac arrest and early-onset AF in the *DES* mutation carriers. These findings suggest an epistatic interaction of DPE4DIP with the *DES* gene, leading to increased penetrance of both traits and AF promotion [47].

**Table 1 cells-11-00416-t001:** Overview of cytoskeletal (-associated) protein variants, identified with whole genome sequencing, that relate to genetic AF.

Protein	Gene	Patho-Mechanism:		Main Conclusions	Refs
		Electrical	Molecular/Functional		
Desmin	*DES*	Changes AERP	Protein aggregates, PQC, autophagy.	60% of patients exhibit conduction disease and arrhythmias, of which 9% is AF.	[32]
				Arrhythmogenic, not related to connexin redistribution.	[48]
Lamin A/C	*LMNA*	↓ I_Na_	PQC, HSP, myolysis, nuclear blebbing, nuclear protein aggregation, altered nucleocytoplasmatic transport.	52% of individuals with R331Q experienced AF.	[31]
				50% of individuals with c.475G>T, p.E159* experienced AF.	[49]
				p.R399C associated with AF and lone AF.	[50]
				c.544C>T, p.Q182* associated with PAF drives progression to permanent AF.	[51]
				Nonsense mutation c.G1494A, p.W498* associated with AF.	[52]
Titin	*TTN*	Abnormal ECG	Disruption sarcomeres, fibrosis (zebrafish).	Truncated titin variants associated with AF.	[53]
				Loss of function variants in titin is associated with early-onset AF.	[54]
				Loss of function mutation in titin is related to early-onset AF, including in ethnic minority probands.	[55]
Myosin heavy chain	*MYH6*	NA	Hypertrophy.	Genetic variants of MYH6 associated with AF.	[55]
	*MYH7*			47% of p.R663H mutation carriers with ventricular hypertrophy showed with AF.	[56]
	*MYH7*		Atrial fibrosis, impairment of thick filament assembly.	Patients with p.A1379T gene mutation present extensive atrial fibrosis without clear ventricular involvement.	[57]
Myosin light chain	*MYL4*	NA	Destabilization of F-actin—Z-disk complex.	p.E11K mutation causes early-onset AF.	[58]
				All *MYL4* c.234delC; p.Cys78Trpfs*29 carriers showed early-onset AF.	[59]
Connexin 40, 43	*GJA5*	Electrical conduction changes	LOF gap junction coupling.	Association between AF and connexin 40 p.Q236H, p.K107R, p.L223M, and p.I257L variants. Connexin 40 p.Q236H mutation is linked with impaired gap junction activity.	[60]
				Connexin 40 p.A96S mutation is associated with lower junctional conductance and enhanced sensitivity voltage gating.	[61]
	*GJA1*			Genetic mosaicism of connexin 43 c.932delC variant in AF patient. Connexin 43 c.932delC had negative effect on gap junction formation and function.	[62]
Junctophilin 2	*JPH2*	NA	LOF impaired RyR2 stabilization, spontaneous Ca^2+^ release.	*JPH2* p.E169K mutation is linked with higher AF inducibility via SR Ca^2+^ by RyR2 destabilization.	[63]
Nucleoporin 155	*NUP155*	ECG abnormalities, APD ↓	LOF nuclear localization, loss nuclear permeability for HSP70.	NUP155 p.R391H human mutation associated with AF. Mutant *NUP155* mice presented reduced atrial AP duration and impaired nucleocytoplasmic transport of HSP70.	[64]
Nesprin 2	*SYNE2*	NA	SYNE is involved in RNA polymerase II binding and alternative splicing.	*SYNE2* A + 688G mutation associated with AF.	[65]
Filamin C	*FLNC*	ECG abnormalities	Reduced localization at Z-disk, but preserved at intercalated disk. Diminished contractile activity.	*FLNC* variants are linked with AF. *FLNC* p.V2297M mutation causes cardiomyocyte dysfunction.	[66]

AERP: atrial effective refractory period; AF: atrial fibrillation; PAF: paroxysmal AF; AP: action potential; ECG: electrocardiogram LOF: loss of function; NA: not available.

Interestingly, a recent study identified multiple index patients with DCM carried an identical mutation (c.59926 + 1G>A) in the *TTN* gene, encoding the giant titin protein, which is the major gene underlying inherited DCM. The identified variant likely leads to truncated titan (TTNtv), which is associated with AF onset [34,53]. In an important subset (53%) of index patients and their family members carrying this founder mutation, atrial tachyarrhythmias, in particular AF, were demonstrated. In three patients, AF preceded the development of TTNtv-associated DCM by 11–14 years [34]. In patients with TTNtv-associated DCM and AF, left atrial enlargement was not noticed in half of them. In addition, the traditional risk factors for AF were absent, suggesting a potential intrinsic effect of TTNtv [34], which is in line with the study of Ahlberg et al. [53]. Together, these data strongly suggest that (paroxysmal) AF is an important part of the clinical disease spectrum caused by TTNtv, even if major structural heart abnormalities are still absent, and the individual does not present traditional risk factors for AF. However, the underlying mechanism of TTNtv causing cardiomyocyte structural and contractile remodeling and, consequently, AF remains to be elucidated.

Thus, emerging evidence has identified novel variants in cytoskeletal proteins to underlie early AF (Table 1). Currently, no detailed mechanistic explanation, treatment modalities, or diagnostic screening tools for genetic AF are available. To overcome this, we first need to understand the mutation-specific molecular pathways that drive genetic AF. This knowledge may ultimately lead to the identification of novel drug targets and therapeutic and diagnostic strategies.

## 4. Pathophysiological Mechanisms Driving Cytoskeletal Protein Mutation-Induced AF

As mentioned previously, all cytoskeletal proteins are linked via the microtubule network with the outer cell membrane, the sarcomeric proteins, and the nuclear membrane, and thereby regulate sarcomere architecture and function as well as nuclear morphology, DNA stability, and gene expression (Figure 2) [19,67,68]. The malfunction of cytoskeletal proteins has been associated with cardiac manifestations such as compromised conduction disorders and arrhythmogenesis and, as such, contributed to clinical phenotypes compatible with DCM as well as AF [69]. Knowledge of the molecular mechanisms underlying cytoskeletal protein mutation-induced AF is needed to develop therapeutic and diagnostic tools for these patients. Although research findings provide some preliminary insights into underlying mechanisms, as, for example, TTNtv, as described above, most findings are still in their infancy (Figure 3).

### 4.1. Desmin Variant-Induced Remodeling

Desmin is one of the intermediate filaments that integrates the sarcolemma, Z disk, and nuclear membrane in sarcomeres and regulates sarcomere architecture. Desmin is encoded by the *DES* gene and highly expressed in the heart, the conduction system of the heart, and the pulmonary vein myocardial sleeve, which is the original focus of AF.

A previous study in desmin knockout mice showed a reduction in the atrial refractory period and increased susceptibility for AF [48], suggesting a key role of desmin dysfunction as a molecular substrate for the onset of AF. Isolated sinoatrial nodes from *DES*^−/−^ mice showed a significant increase in pacemaker potentials and diastolic depolarizations. Furthermore, these mice showed increased PR intervals and P wave durations combined with supraventricular premature beats, which may trigger AF onset [70].

The role of desmin in AF promotion has been confirmed by various additional studies. In iPSC-derived cardiomyocytes expressing the homozygous mutation *DES* p.Y122H [71] and the myocardial tissue of patients carrying *DES* p.A120D [72], impaired assembly of the intermediate filament structure and the presence of cytosolic desmin aggregates were observed. Importantly, variants in *DES* near the c-terminus of the protein tend not to hamper filament formation; however, they interfere with the mechanical properties of the desmin network [73,74].

Moreover, in a mouse model for desmin-related cardiomyopathy, it was shown that mitochondrial dysfunction preceded the onset of cardiac remodeling and pathological heart function [75]. Here, desmin misfolding suppresses mitochondrial respiration and membrane potential and increases the ADP/ATP ratio and mitochondrial DNA (mtDNA) release, which was accompanied by the disruption of the cardiomyocyte cytoskeletal network [76]. Interestingly, cardiac remodeling was shown to be subject to mitochondrial dysfunction in both experimental models for AF and clinical AF [5].

Furthermore, desmin dysfunction is also linked to cardiomyopathy without being directly linked to a specific desmin mutation. In a cardiomyopathy *Lmna^H222P/H222P^* mouse model, desmin organization was disrupted and accumulated in the cytoplasm. Moreover, desmoplakin, plakoglobin, and connexin-43 were mislocalized, and mitochondrial function was impaired. Interestingly, all the effects were ameliorated by the cardiac-specific over-expression of αB-crystallin [77].

The depletion of desmin in ventricular cardiomyocytes derived from a rat leads to nuclear involution, the infolding of the nuclear lamina, and DNA damage driven by the microtubule network, as well as loss in contractile function [78]. A similar phenotype is seen when depleting the desmin binding partner nesprin 3, which links desmin to the nucleus [78]. This points to desmin as an important protein in the linker of nucleoskeleton and cytoskeleton (LINC) complex and for the maintenance of a proper nuclear shape in cardiomyocytes. This notion was further supported by the rescue of nuclear shape in desmin knockout rats by disrupting the connection between nesprins and Sad1p/UNC84 (SUN) domain-containing proteins via dominant-negative KASH (Klarsicht, ANC1, syne homology) peptides (DN-KASH) [78].

Thus, the findings indicate a key role of desmin dysfunction in AF onset. Desmin dysfunction may result in desmin aggregate formation, mitochondrial dysfunction, and disruption of the cytoskeletal network.

### 4.2. Mutations in Lamin A/C

The lamin A/C are type V intermediate filament proteins expressed in terminally differentiated somatic cells, encoded by the *LMNA* gene [79,80]. As lamin A/C are expressed in the nuclear envelope, they provide structural function and transcriptional regulation in the cell nucleus. How *LMNA* variants cause AF onset is unknown, but experimental data from cardiomyopathy and myopathy studies may identify potential mechanisms.

A recent study utilizing induced pluripotent stem cell-derived cardiomyocytes (iPSC-CM) from DCM patients with an *LMNA* p.E342K variant showed a decrease in the spontaneous contraction rate and I_f_ density, a prolonged APD and increased I_Ca,L_, and an increase in arrhythmias [81]. Ventricular cardiomyocytes isolated from *LMNA*^N195K/N195K^ mice also showed a prolonged APD and exhibited a significant increase in both the peak and late I_Na_ [82]. These experimental findings indicate that *LMNA* variants result in electrophysiological alterations, which may drive AF.

The nuclear lamin A/C regulates transcription factors either by sequestering them to the nuclear periphery or by a mechanism involved in activating specific signaling pathways [83,84]. Furthermore, lamin A/C interacts with chromatin by binding to specific genomic regions defined as lamin-associated domains (LADs) [85]. In a recent study on patient iPSC-derived *LMNA* p.K219T cardiomyocytes, mutant lamin A/C suppressed sodium channel *SCN5A* expression, resulting in diminished I_Na_ and impaired signal propagation [86].

So far, studies in ventricular cardiomyocytes revealed *LMNA* variants to result in cytoskeletal and microtubule disruption [35,37,77], dysmorphology of the nuclei [37], activation of the DNA damage response [87], and PARP1 activation [39,88]. PARP1 activation results in the consumption of mitochondrial NAD^+^ levels [88], which drive cardiomyocyte dysfunction and cardiomyopathy onset [39,89]. In DCM, all these effects were ameliorated by supplementation with a precursor of NAD^+^, nicotinamide [39], or conservation of the cytoskeletal network with geranylgeranylacetone (GGA), a heat shock protein (HSP)-inducer [77]. Importantly, the activation of the DNA damage-PARP1-NAD^+^depetion axis has also been found to drive ‘wear and tear’ AF [90,91], indicating a potential key pathway in *LMNA* mutation-induced AF.

Furthermore, minor deviations on lamin A/C are linked to morphological abnormities of the nucleus that cause increased nuclear fragility and mechanosensitivity [92]. Emerging research findings uncovered that the expressions of the various lamin subtypes are dependent on the degree of mechanical stress. A-type lamins are viscoelastic and, therefore, provide structural stiffness, whereas the elastic B-type lamins allow nuclear deformability [93,94]. Mechanotransduction between the structural components of the cardiomyocyte requires the interaction of lamin A/C with the proteins involved in the LINC complex [95]. Interestingly, the disruption of the LINC complex significantly ameliorates DCM progression, prolongs longevity, and helps to retain nuclear morphology. The findings indicate that the LINC complex plays a detrimental role in nuclear morphology and cardiac disease progression in *LMNA*-compromised mice [96]. Whether *LMNA* variants drive AF due to mechanostress transduction on the nucleus via the LINC complex, to changes in transcription and chromosomal organization, or due to an interplay between both remains to be elucidated.

### 4.3. Variants in Myosin Heavy Chain

The two isoforms of cardiac myosin heavy chain (MYH), α and β, are coded by the *MYH6* and *MYH7* genes, respectively. The atrial myocardium expresses αMYH, and the ventricular myocardium expresses βMYH. *MYH7* gene variants are associated with a variety of cardiomyopathies in humans, including HCM [97] and DCM [98]. Emerging evidence also indicates that *MYH6* variants may result in AF. A large whole-genome sequencing study in Icelanders showed that a missense variant in *MYH6* p.R721W is related to several cardiac diseases, particularly conduction disorders, including AF and sick sinus syndrome [99].

Furthermore, miR-208a is encoded by the intron of *MYH6*, which is a cardiac-specific highly conserved microRNA (miRNA). In mice, miR-208a regulates normal cardiac conduction and is required for the expression of *GJA5* (Cx40), indicating a link between *MYH6* and *GJA5* expression and cardiomyocyte function via miR-208a [100]. Recent experimental findings in *MYH6* knockout zebrafish show the activation of the ER stress pathway and enlargement of the cardiomyocytes [101]. In HL-1 and isolated rat atrial cardiomyocytes, the overexpression of *MYH6* resulted in sarcomere impairment, electrophysiological abnormalities, and a slower conduction velocity [102]. As AF is also associated with all these endpoints, comparable pathways may play a role in mutant *MYH6*-induced AF.

### 4.4. Mutation in Myosin Light Chain

Myosin light-chain 4 (MYL4) is important for the contractile function of the cardiomyocytes. Variants in the *MYL4* gene have been associated with clinical AF and have been further explored in experimental studies [58]. In zebrafish, mutant *MYL4* leads to the disruption of the sarcomeric structure, atrial enlargement, and electrical abnormalities. These changes are also observed in clinical AF. In addition, electron microscopic studies revealed myofibrillar disarray and the disappearance of Z-disks, which were related to the destabilization of the F-actin–Z-disk complex, impaired Ca^2+^ signaling, and the onset of atrial myopathy and atrial arrhythmia [58]. Comparable molecular and structural adaptations in the cardiomyocytes may also drive *MYL4* mutation-induced AF.

### 4.5. Mutations in Connexin-40 and Connexin-43

Connexins are channel proteins that form hemichannels specialized in cell-to-cell communication at the intercalated disks [103]. Connexins enable the passive diffusion of compounds up to 1 kDa between two connected cells [104]. Depending on the connexin isoform, they transport water, ions, and second messengers, such as IP3, ADP, ATP, cAMP, and miRNAs [104]. Because of this role, connexins play a vital role in the heart by coordinating the depolarization of cardiac muscle and ensuring cardiac muscle function. The primary atrial connexin proteins are connexin-43 (Cx43) and connexin-40 (Cx40), encoded by *GJA1* and *GJA5*, respectively [105]. Abnormalities in connexin levels, function, or localization impair electrical propagation and can lead to the initiation and/or stabilization of reentry circuits that support AF [105]. Moreover, Cx40 and Cx43 get mislocalized in case of *LMNA* variant-induced cytoskeletal changes [36,77,106,107] and *DES* variants [108]. This observation indicates that impairment in the cytoskeletal network at a specific location affects the network as a whole.

In ‘wear and tear’ AF, reduced Cx40 but not Cx43 levels were found in the atrial tissue of patients with AF compared to controls [109]. As such, several Cx40 mutants have been associated with clinical AF onset [110]. HL-1 atrial cardiomyocytes overexpressing functional Cx40 mutants G38D, V85I, or L229M, resulted in protein instability and accelerated degradation by the proteasome. Similarly, Cx40 mutants cause the gain of the function of hemichannels (half of the gap junction channels) [111,112], resulting in altered conductive and permeability properties [113], which may drive Cx40 mutant-induced AF.

### 4.6. Variants in Junctophilin 2

Junctophilin 2 (*JPH2*) is a member of the junctophilin gene family, which plays an important role in SR Ca^2+^ handling and the modulation of ryanodine receptors (RyRs) [114]. Beavers et al. screened 203 unrelated HCM patients and uncovered the *JPH2* p.E169K variant in two patients with juvenile-onset paroxysmal AF [63]. Follow-up studies in mouse models demonstrated that mice expressing *JPH2* p.E169K exhibited a higher incidence of inducible AF compared to wild-type mice [63]. These changes were attributed to reduced binding of JPH2 p.E169K to RyR2, resulting in abnormal Ca^2+^ release events, which are associated with AF [63]. In addition, atrial JPH2 protein levels in mice correlated negatively with the incidence of pacing-induced AF, indicating a crucial role of JPH2 in atrial cardiomyocyte function and AF promotion [115].

### 4.7. Variants in Nucleoporins

Nucleoporins (NUPs) are the critical molecular components for the assembly and function of the nuclear pore complex (NPC). The NPC is composed of 30 different nucleoporins that are each present in multiple copies [116] and contain cytoplasmic and nucleoplasmic filaments [117]. The filaments consist of microfilaments, intermediate filaments, and myosin filaments, which are all part of the cytoskeletal network. In addition, NPCs interact with the LINC-complex [118]. Due to its localization in the cardiomyocytes, NUPs are cytoskeletal-associated proteins [119].

Alterations in the NPC have been related to a variety of cardiac diseases, such as heart failure [120,121] and AF [64]. The first observation describing the relation between NUP and AF was a study including a large consanguineous AF family from Uruguay and South America, with 57 family members carrying the *NUP155* p.R391H variation [122]. Follow-up studies in mouse models demonstrated that the homozygous R391H variation in *NUP155* resulted in impaired nuclear localization of NUP155, reduced nuclear envelope permeability for HSP70, shortened action potential duration, and AF onset [64]. In addition, *NUP155* is an essential gene, as *NUP155^−/−^* knockout mice died during embryogenesis, whereas heterozygous *NUP155^+/−^* mice were viable and showed reduced action potential durations, which may relate to AF onset [64]. Similarly, a study expressing truncated NUP155 in mouse embryonic stem cell lines showed impaired genome integrity compared to controls, which may also underlie AF [123].

### 4.8. Variants in Nesprin-1/2

Nesprins (*SYNE*) are a family of multi-isomeric scaffolding proteins. Nesprin 1/2 are located at the nuclear envelope and interact with SUN domain-containing proteins, lamin A/C and emerin, to form the LINC complex (Figure 2) [124]. The LINC complex provides a stable physical connection between the nucleus and the cytoskeleton [124]. As mentioned above, the LINC complex participates in many cellular activities, including the maintenance of nuclear morphology, nuclear positioning, the mediation of mechanical cell signaling, and downstream gene regulation. Nesprin 1/2 are highly expressed in skeletal and cardiac muscles. Recent studies showed that variants in *SYNE1* are linked to nuclear morphological changes and the mislocalization of and perturbed interaction with binding partners lamin A/C, emerin, and SUN2 [124,125]. Furthermore, variants in the *SYNE1* and *SYNE2* isoforms are also linked to abnormal mechanotransduction and altered gene expression [126]. These findings indicate that variants in nesprin may result in the derailment of the cytoskeletal network comparable to other mutant LINC-associated proteins, including lamin A/C and, as such, may drive AF.

### 4.9. Variants in Filamin C

Filamin C *(FLNC)* is one of three filamin protein that participates in sarcomere stability [127]. Filamin C is most commonly found in skeletal and cardiac muscles and is localized to the Z-disc, myotendinous junctions, the sarcolemma, and intercalated discs. *FLNC* variants may lead to proteostasis derailment, such as protein misfolding and aggregation and, consequently, the saturation of the ubiquitin–proteasome and autophagy pathways [128]. Since ‘wear and tear’ AF is also associated with proteostasis derailment and autophagic protein degradation, a comparable pathway may drive mutant *FLNC*-induced AF.

## 5. Conclusions

In 15% of AF patients, AF is caused by gene variants. Although emerging research findings indicate that variants in cytoskeletal (-associated) proteins may trigger early AF onset, knowledge on the underlying pathophysiological mechanisms is still in its infancy. Recent findings suggest that cytoskeletal (-associated) protein variants cause disruption of the structural network, mitochondrial dysfunction, electrophysiological changes, DNA damage-induced PARP1 activation, and NAD^+^ depletion. As several of these mechanisms are also found to play a key role in ‘wear and tear’ AF, they may represent canonical pathophysiological pathways driving AF. Future research on cytoskeletal (-associated) protein variants should be directed at the dissection of the molecular root causes driving AF and the identification of novel druggable targets for therapy. As no treatments are available for variant carriers, these studies are clinically highly relevant.

## Figures and Tables

**Figure 1 cells-11-00416-f001:**
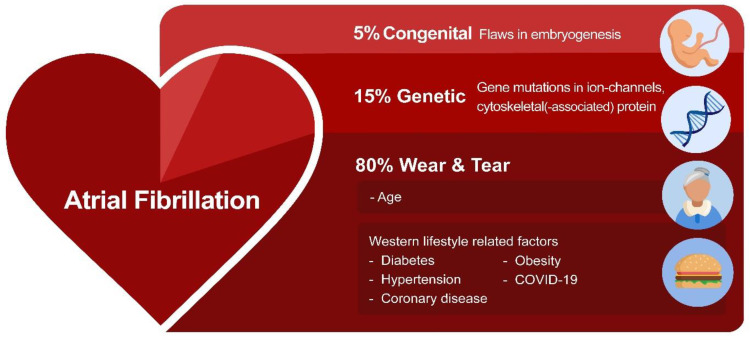
The variety of root causes that drive AF. In most AF patients, environmentally-induced ‘wear and tear’ related to aging or the Western lifestyle triggers AF. In addition, flaws in genetics (variants in ion channels and cytoskeletal proteins) and embryogenesis contribute to AF susceptibility.

**Figure 2 cells-11-00416-f002:**
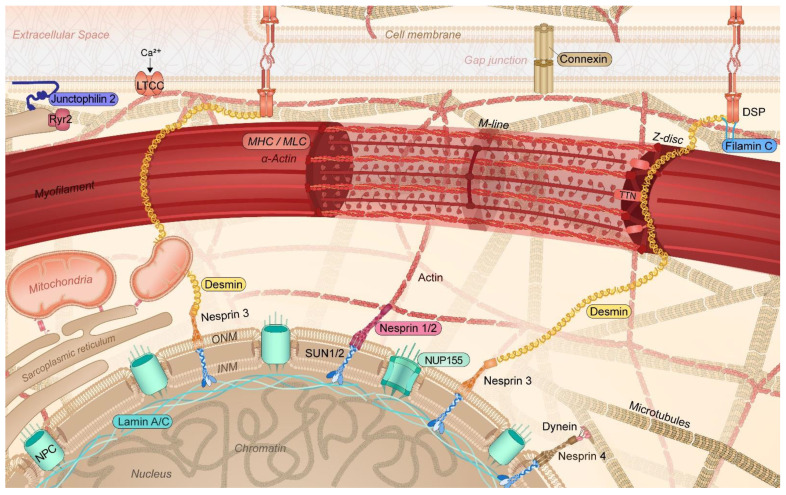
Schematic representation of the specific cytoskeletal protein complexes in cardiomyocytes linked to the onset of AF. Cytoskeletal (-associated) protein variants related to AF are highlighted (colored box). DSP: desmoplakin; INM: inner nuclear membrane; LTCC: L-type calcium channel; MHC: myosin heavy chain; MLC: myosin light chain; NPC: nuclear pore complex; NUP155: nucleoporin 155; Ryr2: ryanodine receptor 2; TTN: titin; ONM: outer nuclear membrane; SUN1/2: Sad1p, UNC-84 domain-containing protein 1/2.

**Figure 3 cells-11-00416-f003:**
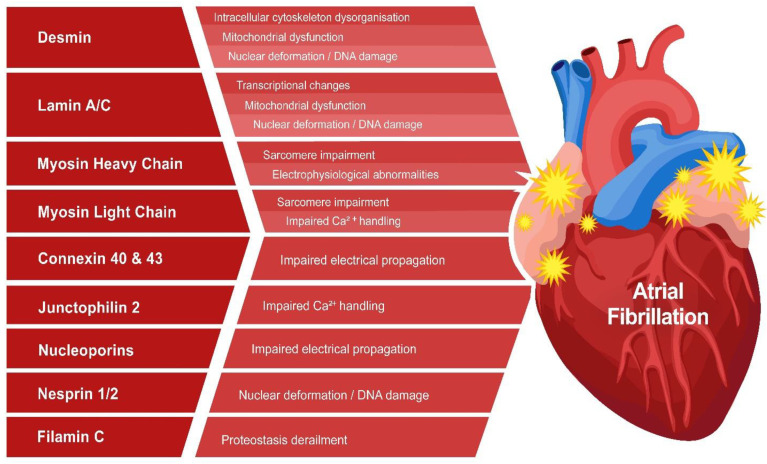
Overview of potential pathophysiological pathways of cytoskeletal (-associated) protein variants driving AF.

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
