# Peer review of "Cytoskeletal Protein Variants Driving Atrial Fibrillation: Potential Mechanisms of Action"

_cells, 2022, doi:10.3390/cells11030416_

Round 1

Reviewer 1 Report

In this article, van WijK and collaborators aim at reviewing the actual knowledge linking cytoskeletal protein mutation and the development of atrial fibrillation. They further discuss the potential mechanisms involved in this process.

In this article, the authors provide the reader with a first overview of atrial fibrillation prevalence, comorbidities and of the complexity of the clinical substrate. They further discuss the importance of the cytoskeletal scaffolding matrix for the cardiomyocyte function. In a third chapter, they are reviewing the distinct mutations identified in a subset of cytoskeletal protein and give insight into their association with clinical atrial fibrillation. Subsequently they provide the reader with a point-by-point discussion of the potential mechanism involved. Their conclusion being that little is known about the mechanisms linking cytoskeletal proteins impairment in atrial cardiomyocytes and the development of atrial fibrillation substrate.

My comments are delineated in the following:

The review article is of interest as it raises awareness on the importance of the integrity of the cytoskeletal architecture to ensure a proper function of atrial cardiomyocytes. In addition, such review is usually focusing on ventricle, to my knowledge this is the first addressing the association with atrial fibrillation.

Despite, this keen interest, I believe that the actual manuscript would benefit from several modifications.

  • Connexins are ion channels and therefore they cannot be listed as cytoskeletal proteins. Information related to connexins should be removed from the manuscript (i.e., table and dedicated chapter). This will give more focus to the article.

  • In the Figure 1: In the section Wear & Tear, I would suggest replacing “Western lifestyle related factors” by “Western lifestyle risk factors”. Furthermore, obesity is missing in this list despite being mentioned in the introduction. Please add.

  • The Figure 2 gives an excellent overview on how the cytoskeletal framework is organized within the atrial cardiomyocytes. This constitutes a beautiful illustration of the intricate interactions between the diverse cytoskeletal proteins. In this direction, I believe that the manuscript would benefit from an additional figure reporting a schematic structure of each of the reported cytoskeletal protein, with indication on their specific features (kinase domains, binding sites, …) and the localization of the principal mutation points discussed within the review article. This would really strengthen the highlights of manuscript.

  • Of note, line 233: SCN5A encode for the expression of a sodium channels not a sodium transporter. Please correct accordingly.    

  • An attention to editing details should be given. For instance, there is two chapters 3, many typo mistakes were also identified. I will not list these as they are too numerous to mention. Please revise the manuscript accordingly.

  • I would replace “presented with AF” with either “exhibited AF” or “experienced AF” or “shown AF”. Also, “Ameliorated” could be replace by “improved” for instance.

Author Response

Reviewer 1:

We like to thank reviewer 1 for his/her positive comments and feedback, which helped to improve the quality of our manuscript. We have modified the manuscript as suggested. The specific changes made are described below. All changes in the text of the manuscript are provided as tracked changes.

R1: Connexins are ion channels and therefore they cannot be listed as cytoskeletal proteins. Information related to connexins should be removed from the manuscript (i.e., table and dedicated chapter). This will give more focus to the article.

We agree that the function of connexins include ion channel,  but their function is different compared to classical ion channels, including sodium, potassium and calcium channels. Latest are involved in membrane currents that generate an action potential. Connexins are present specifically in the gap junctions and composed of two hemi-channels, each with six connexin subunits. They enable passive diffusion of compounds up to 1 kDa between two connected cells and therefore are rather unspecific (PMID 33922534). Depending on the connexin isoform it can transport H2O, moste ions, second messengers such as IP3, ADP, ATP, cAMP, and micro RNAs (PMID 33922534). As such, their function in the cardiomyocyte is different compared to the classical ion channels.

As connexins are associated with the microtubule network (PMID 11553331), description of connexins as ‘cytoskeletal-associated proteins’ is appropriate. Furthermore, variants in connexins are correlated with AF, and impairment of the cytoskeletal network may represent an underlying pathophysiological pathway.

We added information on the function of connexins to the text (line 314):

‘Connexins enable passive diffusion of compounds up to 1 kDa between two connected cells [104]. Depending on the connexin isoform they transport water, ions, and second messengers, such as IP3, ADP, ATP, cAMP, and micro RNAs [104]. Because of this role, connexins play a vital role in the heart by coordinating depolarization of cardiac muscle and ensuring cardiac muscle function.’

And line 322:

‘Moreover, Cx40 and Cx43 get mislocalized in case of LMNA variants-induced cytoskeletal changes [36, 77, 106, 107] and DES variants [108]. This observation indicates that impairment in the cytoskeletal network on a specific location effects the network as a whole. ‘

R1: In the Figure 1: In the section Wear & Tear, I would suggest replacing “Western lifestyle related factors” by “Western lifestyle risk factors”. Furthermore, obesity is missing in this list despite being mentioned in the introduction. Please add.

We appreciate these suggestions for improving figure 1. We implemented these suggestions into a revised figure 1.

R1: The Figure 2 gives an excellent overview on how the cytoskeletal framework is organized within the atrial cardiomyocytes. This constitutes a beautiful illustration of the intricate interactions between the diverse cytoskeletal proteins. In this direction, I believe that the manuscript would benefit from an additional figure reporting a schematic structure of each of the reported cytoskeletal protein, with indication on their specific features (kinase domains, binding sites, …) and the localization of the principal mutation points discussed within the review article. This would really strengthen the highlights of manuscript.

We like to thank this reviewer for the compliment on figure 2. We agree that an additional figure with schematic structure of each of the reported cytoskeletal proteins if of interested but to our opinion such a figure will become very complex as for each gene, several variants present in various domains are associated with AF.

To compromise, in a revised Figure 2 and now highlight the names of the proteins variants associated with AF in colored boxes. Furthermore, the pathophysiological mechanism is now depicted in Figure 3.  

R1: Of note, line 233: SCN5A encode for the expression of a sodium channels not a sodium transporter. Please correct accordingly.    

We appreciate that the reviewer pointed out this mistake. We corrected the mistake.

  • Changed ‘transporter’ into ‘channel’

R1: An attention to editing details should be given. For instance, there is two chapters 3, many typo mistakes were also identified. I will not list these as they are too numerous to mention. Please revise the manuscript accordingly.

We thank the reviewer for pointing out minor mistakes. We adapted all minor typos and numerical errors.

R1: I would replace “presented with AF” with either “exhibited AF” or “experienced AF” or “shown AF”. Also, “Ameliorated” could be replace by “improved” for instance.

We appreciate that the reviewer suggestion and we implemented the reviewers suggestion throughout the text.

Reviewer 2 Report

Wijk et al. reviews atrial fibrillation (AF) focusing on familial onset atrial fibrillation that represents 15% of the AF population. The authors discuss variants identified in genes encoding cytoskeletal proteins including Titin, desmin, myosin heavy chain etc. The authors provide an overview on the pathophysiological pathways that lead to the sustained arrhythmia. Overall, the review has major limitations that are listed below. 

Comments:

  1. The authors stated at the beginning of their review that they will use data provided from patient studies obtained through genome and exome sequencing techniques to identify “mutations”. It would be better to use the word “variants” instead of “mutants” since we are referring to patients and this nomenclature needs to be fixed throughout the manuscript.
  2. The authors stated at the beginning that they will focus on cytoskeletal proteins, are connexins cytoskeletal proteins? No, they are transmembrane proteins that form intercellular channels (gap junctions) between plasma membranes of 2 adjacent cells. They are ion channels proteins, why did the authors include them ignoring variants identified in genes encoding Nav1.5, HCN4, potassium channel proteins and others?
  3. Is the percentage of variants identified in genes encoding cytoskeletal proteins higher than variants identified in ion channel genes? From the data listed in the review, it sounds that most of the variants are very rare and some of them are associated with cardiomyopathies and the mechanism of AF is unknown.
  4. As the authors stated in their introduction that the focus of the review is the pathophysiological mechanism of AF, the section covering this part is very superficial and the information is very limited only discussing the importance of proteins in sarcomere function and gene expression. I would expect to see how these variants drive changes in AP and result in arrhythmia/AF.
  5. Nucleoporins not “nucleosporins”, please fix. Nucleoporins form nuclear pore complexes and are considered more of transport or signaling molecules responsible for nucleocytoplasmic transport rather than cytoskeletal proteins. Please comment.
  6. There are no future implications on how to use the information provided in the review in future studies and patient management. The section of conclusions needs to be expanded to include future directions.
  7. It would be more helpful in Figure 2 if the pathophysiological mechanism leading to AF is represented diagrammatically in relation to the cytoskeletal proteins.
  8. Gene regulatory network is a novel concept introduced in the molecular basis of familial AF pathophysiology and is not discussed or mentioned in the review. PMID: 32717172
  9. Figure 1 should also include COVID-19 since there is a lot of literature describing/addressing the link between COVID-19 (inflammation) and AF pathogenesis.

Author Response

Reviewer 2:

We like to thank reviewer 2 for his/her positive comments and feedback, which helped to improve the quality of our manuscript. We have modified the manuscript as suggested. The specific changes made are described below. All changes in the text of the manuscript are provided as tracked changes.

R2: The authors stated at the beginning of their review that they will use data provided from patient studies obtained through genome and exome sequencing techniques to identify “mutations”. It would be better to use the word “variants” instead of “mutants” since we are referring to patients and this nomenclature needs to be fixed throughout the manuscript.

We would like to thank the reviewer for this valuable suggestion. We corrected the manuscript accordingly.

  • Changed “mutation(s)” into “variant(s)”

R2: The authors stated at the beginning that they will focus on cytoskeletal proteins, are connexins cytoskeletal proteins? No, they are transmembrane proteins that form intercellular channels (gap junctions) between plasma membranes of 2 adjacent cells. They are ion channels proteins, why did the authors include them ignoring variants identified in genes encoding Nav1.5, HCN4, potassium channel proteins and others?

We agree that connexins are ion channels but their function is different compared to classical ion channels, including sodium, potassium and calcium channels. Latest are involved in membrane currents that generate an action potential. Connexins are present specifically in the gap junctions and composed of two hemi-channels, each with six connexin subunits. They enable passive diffusion of compounds up to 1 kDa between two connected cells and therefore are rather unspecific (PMID 33922534). Depending on the connexin isoform it can transport H2O, most ions, second messengers such as IP3, ADP, ATP, cAMP, and micro RNAs (PMID 33922534). As such, their function in the cardiomyocyte is different compared to the classical ion channels.

As connexins are associated with the microtubule network (PMID 11553331), description of connexins as ‘cytoskeletal-associated proteins’ is appropriate. Furthermore, variants in connexins are correlated with AF, and impairment of the cytoskeletal network may represent an underlying pathophysiological pathway.

We added information on the function of connexins to the text (line 314):

‘Connexins enable passive diffusion of compounds up to 1 kDa between two connected cells [104]. Depending on the connexin isoform they transport water, ions, and second messengers, such as IP3, ADP, ATP, cAMP, and micro RNAs [104]. Because of this role, connexins play a vital role in the heart by coordinating depolarization of cardiac muscle and ensuring cardiac muscle function.’

And line 322:

‘Moreover, Cx40 and Cx43 get mislocalized in case of LMNA variants-induced cytoskeletal changes [36, 77, 106, 107] and DES variants [108].’

R2: Is the percentage of variants identified in genes encoding cytoskeletal proteins higher than variants identified in ion channel genes? From the data listed in the review, it sounds that most of the variants are very rare and some of them are associated with cardiomyopathies and the mechanism of AF is unknown.

We did not make a comparison between the percentage variants in cytoskeletal proteins vs ion-channel proteins. As mentioned in the introduction, a significant amount of patients carrying a cytoskeletal protein variant develop AF and therefore the current review is incentive for future studies.

See Line 117:

‘Over 50% of LMNA [31, 42, 44], 60% DES [32] and 30-60% of TTN [34] mutant-carriers develop arrhythmias and atrial tachycardia, including AF, and these patients reveal a more progressive form of cardiac disease with poor outcome.’

Line 405:

‘Future research on cytoskeletal (-associated) protein variants should be directed at dissec-tion of the molecular root causes driving AF and identification of novel druggable targets for therapy. As no treatments are available for variant carriers, these studies are clinically highly relevant.’ 

R2: As the authors stated in their introduction that the focus of the review is the pathophysiological mechanism of AF, the section covering this part is very superficial and the information is very limited only discussing the importance of proteins in sarcomere function and gene expression. I would expect to see how these variants drive changes in AP and result in arrhythmia/AF.

We would like to thank the reviewer for addressing his/herr concern about the superficiality of the section covering the pathophysiological mechanism. This review is limited by the (current) lack of research regarding molecular mechanisms describing the role cytoskeletal variants in AF onset.

Nevertheless, we updated the sections related to pathophysiological mechanisms (including results on electrophysiological mechanisms; see section 3.1, 3.2, 3.3, 3.5, 3.7) and  added a new figure (Figure 3) describing the potential underlying mechanisms.

R2: Nucleoporins not “nucleosporins”, please fix. Nucleoporins form nuclear pore complexes and are considered more of transport or signaling molecules responsible for nucleocytoplasmic transport rather than cytoskeletal proteins. Please comment.

Nuclear pore complexes are assembled from various nucleoporins, and contain cytoplasmic filaments and nucleoplasmic filaments (PMID 26967283). The filaments consists of microfilaments, intermediate filaments and myosin filaments, which are all part of the cytoskeletal network. As such, nucleoporins are cytoskeletal-associated proteins (PMID 31354529).

We added this information to the manuscript (Line 350).

‘Nucleoporins (NUP) are the critical molecular components for the assembly and function of the nuclear pore complex (NPC). The NPC are composed of 30 different nucleoporins that are each present in multiple copies [116], and contain cytoplasmic and nucleoplasmic filaments  [117]. The filaments consists of microfilaments, intermediate filaments and myosin filaments, which are all part of the cytoskeletal network. Also NPCs interact with the LINC-complex [118]. Due to its localization in the cardiomyocytes, NUP are cytoskeletal-associated proteins [119].

Alterations in the NPC have been related to a variety of cardiac diseases, such as heart failure [120, 121] and AF [64]. The first observation describing the relation between NUP and  AF, was a study including a large consanguineous AF family from Uruguay and South America with 57 family members carrying the  NUP155 p.R391H variation [122]. …’

R2: There are no future implications on how to use the information provided in the review in future studies and patient management. The section of conclusions needs to be expanded to include future directions.

We added to the conclusion section (line: 408):

‘… Future research on cytoskeletal (-associated) protein variants should  be directed at dissection of the molecular root causes driving AF and identification of novel druggable targets for therapy. As no treatments are available for variant carriers, these studies are clinically highly relevant.’

R2: It would be more helpful in Figure 2 if the pathophysiological mechanism leading to AF is represented diagrammatically in relation to the cytoskeletal proteins.

We agree. We improved figure 2 by highlighting the gene variants related to AF in colored text boxes, as also suggested by reviewer 1. To present the pathophysiological mechanisms, we made a new figure (Figure 3).

R2: Gene regulatory network is a novel concept introduced in the molecular basis of familial AF pathophysiology and is not discussed or mentioned in the review. PMID: 32717172

Indeed, gene regulatory networks are a novel concept introduced in AF pathophysiology. Unfortunately, this topic is beyond the scope of the current paper and therefore not included.

R2: Figure 1 should also include COVID-19 since there is a lot of literature describing/addressing the link between COVID-19 (inflammation) and AF pathogenesis.

We implemented the reviewers suggested changes into figure 1. Thank you for this suggestion.

Round 2

Reviewer 1 Report

No additional comments with the exception of a small typo line 146, "titan" should be "Titin"

Reviewer 2 Report

Authors have addressed concerns.